# Evaluation of Antibody Response to Heterologous Prime–Boost Vaccination with ChAdOx1 nCoV-19 and BNT162b2: An Observational Study

**DOI:** 10.3390/vaccines9121478

**Published:** 2021-12-14

**Authors:** Davide Firinu, Andrea Perra, Marcello Campagna, Roberto Littera, Federico Meloni, Francesca Sedda, Maria Conti, Giulia Costanzo, Monica Erbi, Gianmario Usai, Carlotta Locci, Mauro Giovanni Carta, Riccardo Cappai, Germano Orrù, Stefano Del Giacco, Ferdinando Coghe, Luchino Chessa

**Affiliations:** 1Department of Medical Sciences and Public Health, University of Cagliari, 09100 Cagliari, Italy; mam.campagna@gmail.com (M.C.); federicomeloni@hotmail.it (F.M.); mariaconti14@gmail.com (M.C.); giuliacostanzo14@gmail.com (G.C.); momo_erbi@tiscali.it (M.E.); gianmariousaimed@outlook.com (G.U.); loccicarlotta@gmail.com (C.L.); mgcarta@tiscali.it (M.G.C.); gerorru@gmail.com (G.O.); delgiacco@unica.it (S.D.G.); luchinochessa@unica.it (L.C.); 2Department of Biomedical Sciences, University of Cagliari, 09100 Cagliari, Italy; andrea.perra@unica.it (A.P.); francisedda@hotmail.com (F.S.); 3Associazione per l’Avanzamento della Ricerca per i Trapianti O.d.V., Non Profit Organisation, 09100 Cagliari, Italy; roby.litter@gmail.com; 4Medical Genetics, Department of Medical Sciences and Public Health, University of Cagliari, 09100 Cagliari, Italy; 5Laboratory Clinical Chemical Analysis and Microbiology, University Hospital of Cagliari, 09042 Cagliari, Italy; r.cappai@aoucagliari.it (R.C.); fcoghe@aoucagliari.it (F.C.)

**Keywords:** COVID-19, vaccination, immunogenicity, heterologous, ChAdOx1, BNT162b2

## Abstract

In several countries, thrombotic events after vaccination with ChAdOx1 nCoV-19 have led to heterologous messenger RNA (mRNA) boosting. We tested the antibody response to SARS-CoV-2 spike protein four weeks after heterologous priming with the ChAdOx1 (ChAd) vector vaccine followed by boosting with BNT162b2(ChAd/BNT), comparing data of homologous regimen (BNT/BNT, ChAd/ChAd) subjects positive for SARS-CoV-2 after the first dose of BNT162b2 (BNT1dose/CoV2) and convalescent COVID-19. Methods: healthy subjects naïve for SARS-CoV-2 infection were assessed for serum IgG anti-S-RBD response 21 days after priming (T1), 4 (T_FULL_) and 15 (T_15W_) weeks after booster dose. Results: The median IgG anti-S-RBD levels at T_FULL_ of Chad/BNT group were significantly higher than the BNT/BNT group and ChAd/ChAd. Those of BNT/BNT group were significantly higher than ChAd/ChAd. IgG anti-S-RBD of BNT1dose/CoV2 group were similar to BNT/BNT, ChAd/BNT and ChAd/Chad group. The levels among COVID-19 convalescents were significantly lower than ChAd/BNT, BNT/BNT, ChAd/Chad and BNT1dose/CoV2. The proportion of subjects reaching an anti-S-RBD titer >75 AU/mL, correlated with high neutralizing titer, was 94% in ChAd/BNT and BNT/BNT, 60% in BNT1dose/CoV2, 25% in ChAd/ChAd and 4.2% in convalescents. At T_15W_ the titer of ChAd/BNT was still significantly higher than other vaccine schedules, while the anti-S-RBD decline was reduced for ChAd/ChAd and similar for other combinations. Conclusion: Our data highlight the magnitude of IgG anti-S-RBD response in ChAd/BNT dosing, supporting the current national guidelines for heterologous boosting

## 1. Introduction

To alleviate the dramatic consequences of the ongoing SARS-CoV-2 pandemic, there is need for detailed and reliable data on the epidemiology, natural history and treatment possibilities of SARS-CoV-2 infection, as well as about vaccine response. Assessment and post-vaccine monitoring of anti-SARS-CoV-2 antibody specifically targeting and thereby inactivating the spike protein and/or its receptor binding domain (RBD) may provide information about the humoral response and its duration [1].

RNA-based COVID-19 vaccines have been the first to be approved globally, produced by Pfizer and Moderna [2]. The Pfizer vaccine BNT162b2 (BNT) induces neutralizing antibody response and generation of CD8+ and CD4+ T-cells for RBD [3].

The AstraZeneca recombinant ChAdOx1-S COVID-19 vaccine (ChAd) and the Johnson & Johnson Ad.26.COV2.S use replication-deficient modified adenoviral vectors encoding the Spike (S) glycoprotein [4,5]. The occurrence of the rare vaccine-induced immune thrombosis with thrombocytopenia syndrome after ChAd or Ad.26.COV2.S vaccine, with younger women showing the highest risk [6], rapidly led—among countries involved in mass vaccination campaigns—to different recommendations including boosting with an mRNA-based vaccine within 8–12 weeks (e.g., subjects below 60 years of age in Italy) [7]. This was supported by two trials [8,9]. Furthermore, the current emergence and spread of SARS-CoV-2 variants is leading to the assessment of the potential of vaccine combinations on efficacy, safety and to overcome limitations linked to worldwide dose availability, costs and vaccine hesitancy [10].

However, there is scant available data on real-world cohorts of vaccinated subjects primed with ChAd and boosted with mRNA vaccine. In a prospective observational study focused on COVID-19 infection and vaccination (CORIMUN study), we tested the antibody response to SARS-CoV-2 spike protein at 4 and 15 weeks after the second dose with homologous or heterologous vaccine combination.

## 2. Materials and Methods

### 2.1. Subjects Enrolment

Among the CORIMUN study cohort, we enrolled consecutive subjects belonging to the following three groups: post-graduate medical and nursing trainees that received either a heterologous COVID-19 vaccination scheme using ChAdOx1 nCoV-19 as prime and BNT162b2 mRNA as booster 8 to 12 weeks apart (ChAd/BNT), or homologous ChAdOx1 nCoV-19 8 to 12 weeks apart (ChAd/ChAd), or BNT162b2 3 weeks apart (BNT/BNT) vaccination regimens. We also recruited a group of subjects that received the first dose of BNT162b2 vaccine and then tested positive for SARS-CoV-2, which prevented receiving the vaccine booster in accordance with Italian guidelines (BNT1dose/CoV2), and a group of subjects convalescent from symptomatic non-severe COVID-19 (COVID-19 convalescent).

Subjects of the first three groups were naïve to SARS-CoV-2 infection (testing since 2020 gave repeatedly negative rt-PCR test for SARS-CoV-2 and undetectable N-protein IgG at vaccination at T0).

All subjects were matched for age and gender and belonged to a well-characterized cohort without underlying systemic or immune-mediated diseases, no evidence of immunodeficiency, lymphoproliferative disorders, or taking relevant medications. Baseline characteristics of groups are shown in Table 1.

### 2.2. Sample Collection and Storing 

A sample of peripheral blood (10 mL) was obtained by venopuncture immediately before each vaccine dose, and defined as: T0, before first dose; T1, just before the second vaccine shot or +21 days from T0); TFULL, 4 weeks after the booster dose (full vaccination course); T15W, 15 weeks after the booster. The serum was separated by centrifugation (2000× *g* for 15 min) within 3 h of collection and aliquots were stored at −80 °C until use.

### 2.3. Serological Analysis 

Prior to vaccination, we screened subjects of both IgM and IgG against SARS-Cov-2 S-protein and N-protein on Maglumi platform (Snibe, Shenzhen, China), with manufacturer IgM cut-off of 1.0 AU/mL, and 1.1 AU/mL for IgG by a chemiluminescent analytical system (CLIA). Subjects were also asked if they had a positive PCR test in the past and they were cross-matched with the rt-PCR database at the laboratory and hospital records.

The antibody response to S-protein (primary endpoint) was detected with the anti-SARS-CoV-2 S-RBD IgG (Snibe Diagnostics, New Industries Biomedical Engineering Co., Ltd., Shenzhen, China) on a MAGLUMI analyzer. Analytical and clinical features of the assay, including the correlation with viral neutralization assay using the reference plaque reduction neutralization test (PRNT) of 50 titer have been previously published.

### 2.4. Statistical Analysis

Patient characteristics were summarized using means, medians, standard deviations, ranges, and percentages as appropriate. Chi squared tests of independence and Fischer’s exact tests were used for categorical data. Mann–Whitney U and Kruskal–Wallis tests were used for unpaired continuous data. All reported *p*-values represent 2-tailed tests, with *p* ≤ 0.05 considered statistically significant. All variables were analyzed using SPSS. 

### 2.5. Ethical Aspects

Patients were recruited and enrolled in the study protocol at the Teaching Hospital of the Cagliari University. Written informed consent was obtained from all patients and controls in accordance with the ethical standards (institutional and national) of the local human research committee. The study protocol, including informed consent procedures, conforms to the ethical guidelines of the Declaration of Helsinki and was approved by the Ethics Committee of the Cagliari University Hospital on 27 May 2020; protocol number GT/2020/10894 and extension approved 27 January 2021. 

## 3. Results

Among subjects vaccinated and naïve to SARS-CoV-2 infection, 50 participants were included in the BNT/BNT group, 36 in the ChAd/ChAd group, 49 in the ChAd/BNT group. There were only Caucasian subjects enrolled in this study. Subjects vaccinated with a single dose before having SARS-CoV-2 infection (BNT1dose/CoV2) were 15, and convalescent unvaccinated COVID-19 subjects were 24 (analysis time after SARS-CoV-2 positivity median 108 days, IQR 46.25). There was no difference in age, gender, BMI and cigarette smoking comparing the study groups (Kruskall–Wallis χ^2^, *p* > 0.05).

At T0, the screening with Maglumi assay (able to capture response to previous SARS-CoV-2 infection) gave results below the cut-off of 1.0 AU/mL for IgM for all subjects enrolled in the study; IgG above 1.1 AU/mL were detectable only in the COVID-19 convalescent, with a median of 7.10 AU/mL (IQR 10.92).

All subjects that received the vaccine course developed a positive antibody response (defined as >1 AU/mL or higher) at both T1 and TFULL.

### 3.1. Anti S-RBD Antibody Levels 4 Weeks after the Second Vaccine Dose

The median IgG anti-S-RBD levels and IQR at TFULL for each group (vaccinated or convalescent) are presented in Table 2 and Appendix A.

The median IgG anti-S-RBD levels at TFULL (4 weeks after vaccine booster) of ChAd/BNT group were significantly higher than the BNT/BNT group and ChAd/ChAd (Figure 1). Those of BNT/BNT group were statistically significantly higher than ChAd/ChAd (Mann–Whitney test for all comparisons). The median levels of IgG anti-S-RBD of BNT1dose/CoV2 group were similar to those of BNT/BNT, ChAd/BNT and ChAd/ChAd group (*p* = 0.175).

The levels among group of COVID-19 convalescent were significantly lower than BNT/BNT, ChAd/BNT, ChAd/ChAd (*p* = 0.007) and BNT1dose/CoV2 (*p* = 0.039).

Median values of S-RBD-protein IgG were slightly higher in female vs. male subjects across all groups, being statistically significant only in the ChAd/ChAd (51.73 AU/mL vs. 19.89 AU/mL, *p* = 0.015).

The proportion of subjects reaching an anti-S-RBD titer over 75 AU/mL was 94% in BNT/BNT, 25% in ChAd/ChAd, 93.9% in ChAd/BNT, 60% in BNT1dose/CoV2 and 4.2% in convalescent (Chi-square = 100.37, *p* < 0.0001) (Figure 2). 

### 3.2. Anti S-RBD Antibody Levels 3 Weeks after the First Vaccine Dose

At T1, the median IgG anti-S-RBD level in the BNT/BNT group was 46.32 AU/mL (IQR, 83.61), ChAd/ChAd 16.1 AU/mL (IQR, 31.66), ChAd/BNT 26.24 AU/mL (IQR, 31.11), with a significant difference in the BNT/BNT group than ChAd/Chad and ChAd/BNT (*p* = 0.021 and *p* = 0.015, respectively, and *p* = 0.53 for ChAd/ChAd vs. ChAd/BNT).

We evaluated the correlation of anti-S-RBD antibodies at T1, 21 days after the first vaccine dose, and that at TFULL with Spearman’s test. A significant correlation was found only for the BNT/BNT homologous vaccination (r = 0.505, *p* = 0.001).

### 3.3. Anti S-RBD Antibody Levels 15 Weeks after the Vaccine Booster

To obtain data about the short-term evolution of antibody response, we ran an ad-interim analysis of a subgroup of vaccinated subjects already enrolled in the study. The BNT/BNT *n* = 15, ChAd/ChAd *n* = 9, ChAd/BNT *n* = 17 were sampled after booster (T15W) at a median of 15.3, 16.5 and 15 weeks, and the antibody titer declined to 90.73 AU/mL, 19.74 AU/mL and 318.2 AU/mL, respectively (Table 2). All participants, regardless the vaccination schedule, still had detectable anti-S antibodies up to T15W. Dynamics of declining antibody level between 4 and 15 weeks after the booster dose was observed in most of the vaccinees, and on average S-RBD IgG level decreased by 10.87 AU/mL/week (slope m = −0.47) for BNT/BNT, 1.36 AU/mL/week (slope m= −3.80) for ChAd/ChAd, 11.96 AU/mL/week (slope m = −0.43) for ChAd/BNT and 10.86 AU/mL/week (slope m = −0.47) for BNT1dose/CoV2. Of note, a different slope of antibody decay between different vaccination schedules was observed mainly for ChAd/ChAd (Figure 3).

The median IgG anti-S-RBD levels at T15W showed a significantly different decrease vs. TFULL for BNT/BNT and ChAd/BNT (*p* = 0.009 and *p* = 0.001, respectively, Wilcoxon signed-rank test), but not for ChAd/ChAd (*p* = 0.249).

The median IgG anti-S-RBD levels at T15W of ChAd/BNT group were still statistically significantly higher than the BNT/BNT and ChAd/ChAd groups, as well as those of BNT/BNT group with respect to ChAd/ChAd (Mann–Whitney test for all comparisons). 

Median values of IgG were higher in female vs. male subjects across all groups, being statistically significant in the ChAd/BNT (*p* = 0.003) (Appendix A), while in the ChAd/ChAd there were only females.

## 4. Discussion

The ongoing COVID-19 pandemic is also being limited by an unprecedented vaccine program with the innovative use of mRNA and adenovirus-vectored vaccines. Due to limitations in vaccine trials with respect to capturing a quickly evolving scenario, data of short- and long-term response to COVID-19 vaccines is of utmost importance to provide prospective real-life data about immunogenicity.

In many countries, the post-marketing vigilance data for adenovirus-vectored COVID-19 vaccines has led to the emergence of the recommendation to complete the vaccine cycle using a boost with an mRNA-based vaccine within 8–12 weeks [7], on the basis of two trials assessing safety and efficacy of various combinations of heterologous prime-boost vaccination [8,9]. Other studies focused on the ChAd/BNT combination [11,12], interestingly, show antibody neutralizing titers and cellular CD4+ and CD8+ responses equal or higher than ChAd/ChAd or, to some extent, BNT/BNT combination [13].

Furthermore, the ChAd/BNT dosing strategy has been found to generate significantly greater neutralizing titers against the SARS-CoV-2 variants of concern (VOCs) [12,14], including the currently prevalent Delta [15]. 

The present study analyzed the humoral response of a real-world cohort of healthy vaccinated subjects primed with ChAd and boosted with BNT mRNA vaccine compared to homologous dosing (BNT/BNT and ChAd/ChAd). Four weeks after vaccine booster, a time span associated with the maximum antibody response after vaccination [16], the serum IgG anti-S-RBD of the ChAd/BNT group were significantly higher than those of the BNT/BNT group and ChAd/ChAd. Additionally, those of BNT/BNT group were significantly higher than ChAd/ChAd, in accordance with recent studies [11,12]. The IgG anti-S-RBD titers of BNT1dose/CoV2 group were similar to those of the BNT/BNT, ChAd/BNT and ChAd/ChAd groups. The IgG anti-S-RBD levels among the group of COVID-19 convalescents were significantly lower than the heterologous or homologous combinations (BNT/BNT, ChAd/BNT, ChAd/ChAd) and BNT1dose/CoV2, although their decline over time is usually slower than that of vaccinated patients [17,18].

Limitations of the study include the unblinded design and the absence of formal studies with neutralization assays or cellular immunity, as well as the young age and healthy status of the subjects enrolled. The analysis at week 15 after booster was conducted on a limited sample of subjects that reached this timepoint and that were available for blood drawing.

The results of the present study are in line with the estimated overall immunogenicity and efficacy of vaccines assessed in clinical trials with homologous dosing that is related to the peak of antibody titer [19] and highlight the magnitude of IgG anti-S-RBD response in the heterologous ChAd/BNT dosing among a cohort of naïve and well-characterized subjects. Recently, higher antibody titers were observed when comparing healthcare workers that underwent homologous mRNA-1273 compared with those vaccinated with BNT162b2 [20], and in the mRNA-1273 phase 3 trial, the breakthrough infections were associated with lower titers [21].

Previous works demonstrated a strong correlation between serum neutralizing activity using PRNT50 and anti-S-RBD titer to SARS-COV-2 evaluated with the SNIBE assay [22]. An anti-S-RBD titer >75 AU/mL correlates to a high neutralizing titer (≥1:160 PRNT50) for the majority of patients, which is clinically relevant in line with predictive models of protection from SARS-CoV-2 infection and severe COVID-19 disease of vaccine trials [19]. The proportion of subjects reaching an anti-S-RBD titer over 75 AU/mL in our cohort was about 94% in ChAd/BNT and BNT/BNT, 60% in BNT1dose/CoV2, 25% in ChAd/ChAd, and 4.2% in convalescent, and thus in line with previous studies and estimates [22,23]. At 15 weeks, this proportion decreased to 70% for BNT/BNT, 16.7% for ChAd/ChAd, 88.2% for ChAd/BNT, and 20% for BNT1dose/CoV2, and should prompt specific studies about the differences in protection over time against detectable SARS-CoV-2 infection and/or severe COVID-19. Besides the antibody titer, the greater neutralization capacity of the ChAd/BNT dosing against the VOCs, including Beta and Delta and increased T-cell reactivity might be taken into account [13,15].

The evolution of antibody titer over time and outcomes upon exposure to SARS-CoV-2 variants is still unknown for heterologous boost, and here we report novel data on this topic. Anti-spike antibody titer (and the related neutralizing activity) declines over months in most vaccinated subjects with mRNA or adenoviral homologous schedules [24,25], with different rate between subjects previously positive and those naïve to SARS-CoV-2. Overall, the decline from maximum peak to 15th week of 36% observed in ChAd/BNT and 64% in BNT/BNT is slightly higher compared to similar cohorts [24,25], but in line with that observed in cohorts of younger subjects [26,27].

The levels of anti-RBD IgG induced by BNT/BNT and ChAd/ChAd found in our study at 4 and 15 weeks after booster are comparable to those recently reported in cohorts of naïve subjects [26,28]. However, our analysis of ChAd/BNT response shows significant differences at these timepoints. One factor is linked to the higher peak of IgG titer reached after vaccine booster, but the slope between different vaccination schedules indicates a slower decrease in the first 15 weeks mainly for ChAd/ChAd with respect to the heterologous schedules and BNT1dose/CoV2. The small difference in IgG anti-RBD decrease of about 1.1 AU/mL/week might result in significant differences over 6 to 12 months in antibody titer and neutralization, and thus it should be prospectively investigated. The previously demonstrated equal or higher antibody titer for ChAd/BNT combination is still found in our study at more than 3 months after booster, and this interesting finding requires further confirmation and extension to the following months in focused studies, together with an evaluation of efficacy (symptomatic disease, moderate/severe illness). These data may also require further extension to elderly and at-risk subgroups not included here due to the design of our study, such as those that previously showed a higher proportion of low/non-responders to the SARS-CoV-2 vaccines (e.g., those taking immunosuppression, HIV, patients on hemodialysis) [29]. 

Increased risk of hematological and vascular events leading to hospital admission or death were observed for short time intervals after first doses of the ChAdOx1 nCoV-19 and BNT162b2 mRNA vaccines. A recent case–control study that also applied confirmatory self-controlled case series (SCCS) analysis also demonstrated an association between first dose of ChAdOx1 and idiopathic thrombocytopenic purpura (ITP) [30,31].

The risks of most of these events were substantially higher and more prolonged after SARS-CoV-2 infection than after vaccination in the same population [30,32], and that of ITP is overall similar to other vaccines. Furthermore, limitations to the use of adenovirus-vectored COVID-19 vaccines according to the patient’s age have emerged in some countries and triggered the use of heterologous dosing while there was a worldwide shortage of doses and an imbalance in the global distribution, with relevant implications for pandemics control [33,34].

After a thorough analysis of adverse events related to ChAdOx1 and appropriate evaluation of risk-to-benefit ratio, the adenovirus-vectored prime might be boosted by BNT (or other mRNA vaccine) for larger parts of population, assuring a high efficacy of vaccination. This may help to overcome the limitations linked to worldwide dose availability, costs, and vaccine hesitancy [10].

In addition, heterologous vaccination could be of interest for those that may be diagnosed with diseases that would qualify them as extremely vulnerable subjects after the first vaccine dose (requiring mRNA vaccine), or those that may develop severe allergy to polysorbate and require completion of vaccine course [35]. 

Our data highlight the short-term immunogenicity of heterologous dosing, providing data for up to 15 weeks after booster dose, supporting the current national guidelines for boosting with mRNA vaccine after adenoviral-vectored prime. There is need for epidemiological and efficacy studies for these flexible vaccine dosing schemes to understand whether there is a different immune response kinetics with respect to mRNA vaccination at medium and long term, or there is a clinical impact for specific subgroups in order to better tailor vaccination campaign.

## Figures and Tables

**Figure 1 vaccines-09-01478-f001:**
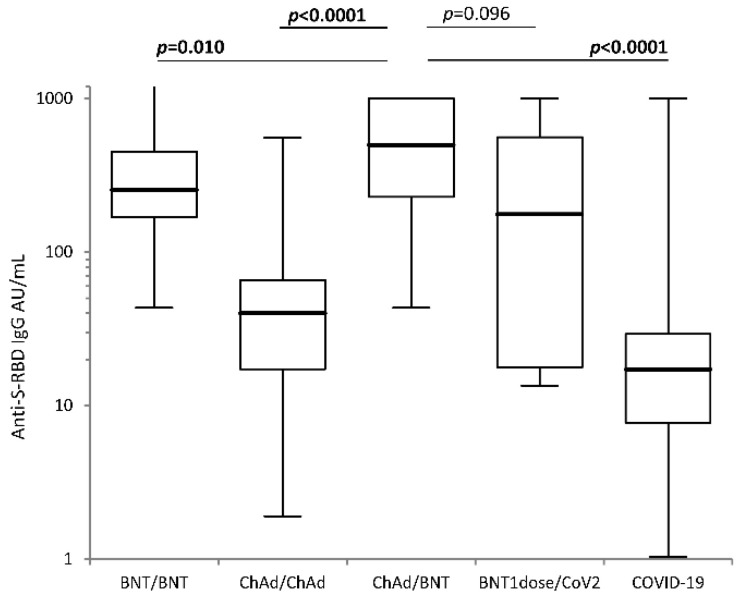
Comparison of SARS-CoV-2 Anti-S-RBD IgG titers between groups of vaccinated and convalescent subjects. Comparison of SARS-CoV-2 serum anti-S-RBD IgG (AU/mL) four weeks after booster (T_FULL_) or in convalescent subjects. BNT/BNT: two doses of BNT162b2; ChAd/ChAd: two doses of ChAdOx1 nCoV-19; ChAd/BNT: first dose ChAd, second dose BNT; BNT1dose/CoV2: BNT first dose, then SARS-CoV-2 infection (no vaccine booster); COVID-19: convalescent.

**Figure 2 vaccines-09-01478-f002:**
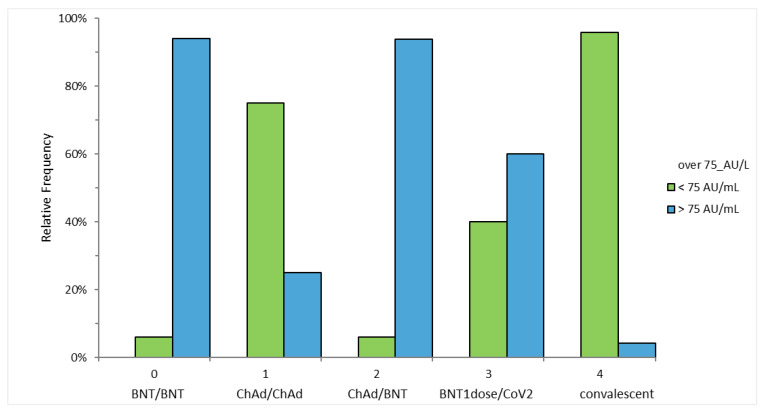
Proportion of subjects reaching an anti-S-RBD titer over 75 AU/mL across study groups.

**Figure 3 vaccines-09-01478-f003:**
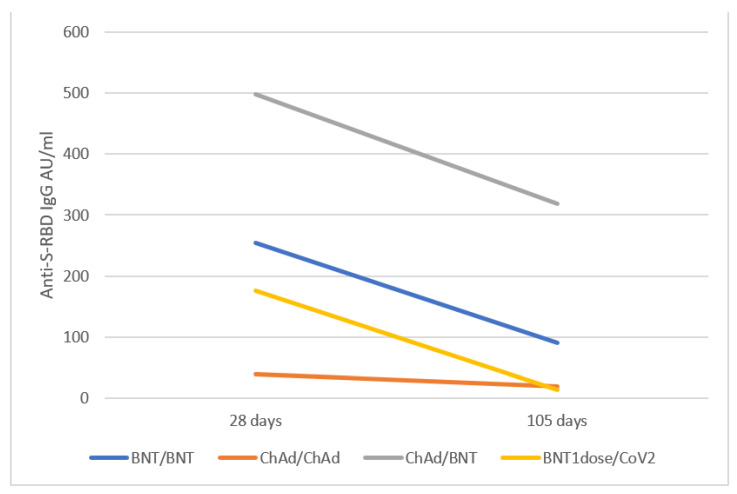
Evolution of SARS-CoV-2 Anti-S-RBD IgG titers between groups of vaccinated subjects. Evolution of SARS-CoV-2 decline of serum anti-S-RBD IgG (AU/mL) at week 4 and 15 after booster. BNT/BNT: two doses of BNT162b2; ChAd/ChAd: two doses of ChAdOx1 nCoV-19; ChAd/BNT: first dose ChAd, second dose BNT; BNT1dose/CoV2: BNT first dose, then SARS-CoV-2 infection (no vaccine booster).

**Table 1 vaccines-09-01478-t001:** Characteristics of enrolled subjects.

Subjects Characteristics	BNT/BNT *n* = 50	ChAd/ChAd *n* = 36	ChAd/BNT *n* = 49	BNT1dose/CoV2 *n* = 15	COVID-19 Convalescent *n* = 24
Age (years), median (IQR)	33 (5.2)	34 (18)	25.2 (18.8)	34 (24)	47 (19.9)
Male, %	42	42.5	19.6	40	33
BMI, median (IQR)	21.75 (5.17)	22.31 (3.88)	20.795 (2.85)	21.42 (4.28)	23.14 (4.42)
Current smoker, %	14	15	15.7	13	7
Diabetes, %	0	0	0	0	0
Weeks between vaccine doses, median (IQR)	3 (0)	11.14 (0.37)	11.5 (0.28)	n.a.	n.a.
Analysis time after vaccine booster (days), median (IQR)	29 (2)	30 (7)	27.5 (7)	n.a.	n.a.

BNT/BNT: two doses of BNT162b2, 3 weeks apart; ChAd/ChAd: two doses of ChAdOx1 nCoV-19, 8 to 12 weeks apart. ChAd/BNT: first dose of ChAdOx1 nCoV-19, second dose BNT162b2, 8 to 12 weeks apart; BNT1dose/CoV2: BNT162b2 first dose, then SARS-CoV-2 infection (no vaccine booster).

**Table 2 vaccines-09-01478-t002:** The IgG anti-S-RBD levels of study groups assessed 4 weeks (T_FULL_) and 15 weeks (T_15W_) after vaccine booster and data of convalescent COVID-19 subjects.

Anti-RBD IgG AU/mL at T_1:_ 3 Weeks after 1st dose	1st Quartile	Median	3rd Quartile	Comparison vs. ChAd/BNT *p*-Value ^§^	Comparison vs. BNT/BNT *p*-Value ^§^
BNT/BNT	18.30	46.32	101.92	0.015	-
ChAd/ChAd	8.72	16.10	40.38	0.534	0.021
ChAd/BNT	10.26	26.24	41.38	-	0.015
anti-RBD IgG AU/mL at T_FULL_: 4 weeks after booster	1st Quartile	Median	3rd Quartile	Comparison vs. ChAd/BNT *p*-value ^§^	Comparison vs. BNT/BNT *p*-value ^§^
BNT/BNT	169.24	253.75	452.15	0.010	-
ChAd/ChAd	17.27	40.015	65.45	<0.0001	<0.0001
ChAd/BNT	228.43	497.60	1000	-	0.010
BNT1dose/CoV2	17.74	176.40	560.13	0.096	0.295
COVID-19 convalescent	7.73	17.19	29.23	<0.0001	<0.0001
T_15W_: 15 weeks after booster	1st Quartile	Median	3rd Quartile	Comparison vs. ChAd/BNT *p*-value ^§^	Comparison vs. BNT/BNT *p*-value ^§^
BNT/BNT	34.00	90.70	109.83	*p* < 0.0001	-
ChAd/ChAd	6.13	19.74	29.26	*p* < 0.0001	0.011
ChAd/BNT	123.00	318.20	400	-	*p* < 0.0001
BNT1dose/CoV2	6.41	13.44	57.97	0.025	0.001

BNT/BNT: two doses of BNT162b2, 3 weeks apart. ChAd/ChAd: two doses of ChAdOx1 nCoV-19, 8 to 12 weeks apart. ChAd/BNT: first dose of ChAdOx1 nCoV-19, second dose BNT162b2, 8 to 12 weeks apart. BNT1dose/CoV2: BNT162b2 first dose, then SARS-CoV-2 infection (no vaccine booster). ^§^ Mann–Whitney test.

## Data Availability

Data and materials are available from corresponding author on reasonable request.

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
