# Peer review of "Evaluation of Antibody Response to Heterologous Prime–Boost Vaccination with ChAdOx1 nCoV-19 and BNT162b2: An Observational Study"

_vaccines, 2021, doi:10.3390/vaccines9121478_

Round 1

Reviewer 1 Report

The authors tested the antibody response to SARS-CoV-2 spike protein 4 weeks (TFULL) after heterologous priming with the ChAdOx1 (ChAd) vector vaccine followed by boosting with BNT162b2(ChAd/BNT) comparing data of homologous regimen (BNT/BNT, ChAd/ChAd), and subjects positive for SARS-CoV-2 after the first dose of BNT162b2 (BNT1dose/CoV2) and convalescent COVID-19. The results showed the median IgG anti-S-RBD levels at TFULL of Chad/BNT group were significantly higher than the BNT/BNT and ChAd/ChAd group and the BNT/BNT group were significantly higher than ChAd/ChAd, supporting the current guidelines for heterologous boosting.

The manuscript is well written and presented, just a few minor points:

  1. Line 154. Need to add a comma between "weeks, and " to be clearer.
  2. Figure 3. Y-axis title is missing. Please add "Y-axis title is missing. Please add "Anti-S-RBD IgG titers in Y-axis".
  3. Line 260. "-64% in BNT/BNT" is it typo?? Please check.

Author Response

We modified the manuscript as detailed below, carefully taking into account the constructive and very useful comments provided.

  1. Line 154. Need to add a comma between "weeks, and " to be clearer.

R: Thanks for the note, we amended the sentence accordingly.

  1. Figure 3. Y-axis title is missing. Please add "Y-axis title is missing. Please add "Anti-S-RBD IgG titers in Y-axis".

R: We added the y Axis title in the figure 3.

  1. Line 260. "-64% in BNT/BNT" is it typo?? Please check.

R: Thank you for the note. The “-“ was removed accordingly.

Reviewer 2 Report

  1. Tables and figures should not be combined into one section, which makes a little hard to connection the text information with the tables and figures. It is better that each of tables and figures closely follow the text.
  2. Table 1 belongs to Materials and Methods and should be moved up to that section.
  3. The authors also evaluated T0 and T1 IgG, but these data are not included in Table 2, although it should be.

Author Response

We modified the manuscript as detailed below, carefully taking into account the constructive and very useful comments provided.

POINT BY POINT REPLY:
Reviewer #2:

  1. Tables and figures should not be combined into one section, which makes a little hard to connection the text information with the tables and figures. It is better that each of tables and figures closely follow the text.

R: We agree with the observation, but this was specified in the instructions for Authors and in the manuscript template. We have now placed tables and figures close to the data in the text of the manuscript. We hope to have improved the clarity and readability. Also, the Editorial office may later take care of layout to further improve clarity.

  1. Table 1 belongs to Materials and Methods and should be moved up to that section.

R: Thanks for the suggestion; we moved the Table 1 to Materials and methods, changing the text accordingly.

  1. The authors also evaluated T0 and T1 IgG, but these data are not included in Table 2, although it should be.

R: The Authors would thank the Reviewer for the suggestion. We added the T1 anti-S-RBD IgG to Table 1. For T0, due the study design and assay characteristics there was data only for COVID-19 convalescent, we added a specific sentence at the beginning of results (lines 130-133 of the revised manuscript).
